# Small Molecular Drug Screening Based on Clinical Therapeutic Effect

**DOI:** 10.3390/molecules27154807

**Published:** 2022-07-27

**Authors:** Cai Zhong, Jiali Ai, Yaxin Yang, Fangyuan Ma, Wei Sun

**Affiliations:** College of Chemical Engineering, Beijing University of Chemical Technology, Beijing 100029, China; 2019200135@mail.buct.edu.cn (C.Z.); 2019400039@mail.buct.edu.cn (J.A.); 2020200163@mail.buct.edu.cn (Y.Y.); 2020400053@mail.buct.edu.cn (F.M.)

**Keywords:** molecular descriptor, molecular fingerprint, Dempster–Shafer theory, Kennard–Stone division

## Abstract

Virtual screening can significantly save experimental time and costs for early drug discovery. Drug multi-classification can speed up virtual screening and quickly predict the most likely class for a drug. In this study, 1019 drug molecules with actual therapeutic effects are collected from multiple databases and documents, and molecular sets are grouped according to therapeutic effect and mechanism of action. Molecular descriptors and molecular fingerprints are obtained through SMILES to quantify molecular structures. After using the Kennard–Stone method to divide the data set, a better combination can be obtained by comparing the combined results of five classification algorithms and a fusion method. Furthermore, for a specific data set, the model with the best performance is used to predict the validation data set. The test set shows that prediction accuracy can reach 0.862 and kappa coefficient can reach 0.808. The highest classification accuracy of the validation set is 0.873. The more reliable molecular set has been found, which could be used to predict potential attributes of unknown drug compounds and even to discover new use for old drugs. We hope this research can provide a reference for virtual screening of multiple classes of drugs at the same time in the future.

## 1. Introduction

The emergence of new diseases (such as COVID-19) and the rise of drug resistance are constantly forcing researchers to discover and develop new drugs with better therapeutic effects and fewer side effects. As an important way to find new drugs, drug development is receiving great attention [1]. It includes a series of procedures, such as the determination of lead compound, clinical trials, and the final review by the National Medical Products Administration [1]. As the early key step, the determination of lead compound is also called drug screening, by which possible candidate drugs to relieve and cure various diseases [2] are discovered and serve as the object for subsequent research. In traditional drug development, this step is conducted through constant experimentation and testing of compounds from small molecule databases, which requires a significant amount of time and money [3]. However, in the last two decades, virtual screening (VS) has gradually become more and more popular [3]. Currently, it plays a significant role in the discovery of small molecule drugs with certain activity [4]. According to different starting points for identifying desirable drugs, vs. can be classified as structure-based virtual screening (SBVS) or ligand-based virtual screening (LBVS). Based on these forms of screening, many drugs with therapeutic effects have been successfully discovered and brought into the market with shorter development period and less investment, such as scopoletin and aliskiren [2,3,4,5].

SBVS identifies a target drug by means of the docking between a target protein and small molecular drugs. In terms of drugs with certain effects, if the three-dimensional structure of a target protein is unknown, the search for corresponding drug molecules cannot be realized based on SBVS [6]. Differing from SBVS, LBVS pays attention to data mining on small molecular drugs based on the assumption that compounds with similar structure have similar properties [7]. All kinds of drug databases have been well established in the history of drug development, which include a huge number of drugs and their structure information [8,9]. With the development of structure digitalization, LBVS methods can accomplish faster computation and predict as many potential candidates as possible. Currently, machine learning technologies further facilitate SBVS and LBVS research [1,10], especially LBVS.

There had been many studies employing various machine learning methods to screen drugs with specific characteristics [11]. Müller et al. used kernel-based classification methods to reduce the error rate of distinguishing drugs and non-drugs [12]. Focusing on clinical trials failure and withdrawal caused by drug-induced liver injury, Li et al. proposed a support vector machine model to identify drugs harmful to the liver [13]. To decrease the failure rate of drug candidates that bind to the androgen receptor, Gupta et al. developed an efficient model to predict their toxicity, particularly focusing on liver injury [14]. For enhancing the hit rate of drugs that are able to treat various diseases by inhibiting the S100A9 target, Lee et al. established predictive models to classify them by applying several classifiers and molecular descriptors [15]. In order to recognize drugs that cause cardiotoxicity by blocking the Kv11.1 channel, Kim et al. applied ensemble models to predict blockers and non-blockers [16]. In these studies, molecular characteristics have been described and classified from different perspectives, i.e., drugs for certain diseases, drugs for certain protein targets, or toxicity induced by certain drugs. In addition, in the study by Lotsch et al., the functional genomics-based criterion is applied to classify drugs from pharmacology, which is suitable for drug classification and provides a phenotypic path for drug discovery and repurposing [17]. Kim et al. constructed a prediction model to provide new indications for herb compounds for certain diseases [18]. It can be seen that most studies have been focused on specific diseases or side effects, which are just properties of drugs. As it is also well known that drugs can be classified by what they treat, such as analgesics, antibacterial agents, and antitumor drugs, common features can be extracted from the same drug category [19]. In the early stage of VS, if drug actual clinical therapeutic effect is taken into account, potential candidates could be screened more quickly and efficiently by direct drug multi-classification [20].

The Anatomical Therapeutic Chemical (ATC) Classification System is formed by hierarchically classifying drugs according to their anatomical, therapeutic, and chemical properties, and many models have been developed to enrich the system [21,22]. Although the studies on the ATC system have included clinical therapeutic effect, the amount of drug in the system is much lower than the actual quantity [21], so the existing drug information has not been fully utilized. Moreover, most prediction models have realized multi-classification for ATC system only by an individual algorithm. At present, no individual classifier can show good classification performance for all data sets. To obtain a more reliable multi-classes prediction model, the fusion method was used to integrate the information of multiple classifiers [23]. According to different objects, fusion methods can be divided into class label fusion, support function fusion, etc. [23]. Compared to the label fusion, the fusion for support functions is more interpretable when similar performance is achieved. As a support function fusion method, Dempster–Shafer (DS) evidence theory has been successfully applied to classification [24]. As Kim et al. provided new repositioning for existing drugs [18], the constructed prediction model identified unknown compounds and discovered new possibilities for marketed drugs through providing prediction probability. DS fusion method can merge the predicted probabilities of multiple classifiers to recognize unknown drugs or even discover new effects of old drugs.

In order to promote drug multi-classification development, seven categories of popular drugs are chosen as examples to obtain data for research. As much as possible, drug molecules with corresponding therapeutic effects are collected to ensure the stability of the model. Differing from simple binary classification, classification for the collected drug molecules is a multi-class prediction issue. There are multi-classification algorithms and multi-classification models combining multiple binary classifiers that can be used to solve the issue. Not only has previous research proved the advantages of binary classifiers, but the applicability of the two-classification strategy to achieve multi-classifications has also been demonstrated by Galar et al. [25].

In this work, a multi-classification method based on DS evidence reasoning theory is proposed to predict possible clinical therapeutic effects of unknown drugs. First, random forest (RF), adaptive boosting trees (ABT), support vector machine (SVM), logistic regression (LR), and linear discriminant analysis (LDA) are selected as the separate classifiers for fusion due to their wide applicability [5,15,16]. Then, based on the DS fusion method, predicted probabilities of the five classifiers are fused into final discriminant probability. The Kennard–Stone (KS) division method [26] is used to divide the data set. The final statistical mean of the test set is used to pick out the model with good performance, which provides more reliable information for potential effects of unknown drugs. Furthermore, the reliability of the obtained model is verified by an external validation set.

The rest of the paper is organized as follows: In Section 2, based on different data sets, the better prediction model is obtained by comparing classification results, and the prediction ability of the model for multi-action drugs is also analyzed through external validation. Section 3 includes data collection and method application. The source of drug molecules, five description data sets for drug molecular structure (represented by the descriptors), and the acquisition process of four molecular sets are introduced in the section on data collection. In method application, five single classifiers, DS fusion principle, measurement method, and indicators for classification performance evaluation are presented. Finally, the conclusion is stated in Section 4.

## 2. Results and Discussion

An appropriate drug molecular set can provide the basis for reliable prediction of unknown drugs. According to comparison of classification results, the suitable molecular set, descriptor set, and classification algorithm are determined. Furthermore, based on the external validation set, the performance of twelve classification models is further compared and verified in Section 2.2.

### 2.1. The Comparison of Different Molecular Sets, Description Sets, and Classification Methods

Based on 240 groups of results obtained from four molecular sets (the acquisition process of four molecular sets is introduced in Section 3.1.3), five descriptor sets, and twelve classification methods, firstly, molecular sets that are suitable to discover the correlation between molecular structure and therapeutic effect are determined. Based on the determined molecular sets, two classification methods are utilized to make a comparison among ten groups of results to discover descriptor sets that are beneficial to characterize the molecular structure. Comparing twelve classification results from a suitable molecular set and descriptor set, the method with good classification performance is obtained.

In Table A2, Table A3, Table A4, Table A5 and Table A6 of the Appendix B, classification results of five descriptor sets are shown in detail. These results are obtained by five individual classifiers and seven fusion methods used on four molecular sets.

#### 2.1.1. The Impact of Molecular Sets

Comparing all results from different molecular sets, the highest Q value, 0.862, is achieved in molecular set S_1_. The highest kappa coefficient, 0.81, is obtained in molecular set S_3_. For the overall results in the five tables, better classification results appear more frequently in molecular set S_1_ and S_4_. The results from data with Mordred descriptors and Morgan fingerprints processed by RF method are shown in Figure 1, with (a) classification performance on Mordred descriptor set and (b) classification performance on Morgan fingerprint set. When the descriptor set is Mordred descriptors, it can be observed that the highest Q value is obtained based on S_1_, and same is true for the largest kappa coefficient. When Morgan fingerprints serve as the descriptor set, RF method performs better on S_4_ with the fewest number of drug molecules, as shown in Figure 1.

It can be noticed that the classification performances of RF are different for the studied data sets. To investigate the root cause for such difference, the content of each data set is not identical from the very beginning, as can be seen in Section 3.1.3. Each data set is constructed by checking multifaceted drug information, such as therapeutic effect, mechanism of action, and phase. Four data sets including different drug molecules are introduced in Section 3.1.3. It can be observed that not all features can be captured effectively for given descriptors, leading to different performance on data sets. On the other hand, the numbers for each drug category are uneven, which could be another reason for the performance difference. This suggests that a better descriptor is important for the structural characterization of drugs. It is also noticeable that molecular sets with better results are inconsistent with two descriptors, which will be discussed in the next section.

#### 2.1.2. The Impact of Descriptor Sets

Comparing the results from different descriptor sets, better classification performance can be achieved by combinatorial descriptors in most cases. Molecular set S_4_ is investigated by single ABT and the classification model fused from RF, SVM, and LR. Corresponding results are shown in Figure 2, where Q and kappa coefficient based on five different descriptor sets can be observed in Figure 2a,b, respectively. Furthermore, aiming at six types of descriptor groups, descriptor information included in five descriptor sets is displayed in Figure 3. Among these groups, “atom-type counts” and “substructure fragments” are representation for zero-dimensional and one-dimensional structure information, respectively, and the remaining four types of descriptor groups are all representation for two-dimensional structure information. Values on the axis for each descriptor represent the number of this type of descriptor in the descriptor set.

Comparing three descriptor sets with only binary value, i.e., MACCS, topological, and Morgan fingerprint data, the results obtained by combinatorial and Mordred descriptors are better, which is attributed to different representation of molecular structure. As shown in Figure 3, there are distinct differences between the two descriptor sets and the other three fingerprint sets in terms of six major types of descriptor groups. The three fingerprint sets are calculated by different principles but all come from digital transformation of molecular one-dimensional structure information, such as substructure fragments. In contrast, the two types of descriptor data composed of real numbers not only contain one-dimensional structure information but also include two-dimensional information, such as topological and connectivity descriptors.

Contrasting classification results of combinatorial and Mordred descriptor sets, although the number of descriptors contained in the former is 755 and less than that of the latter, 1127, the Q value and kappa coefficient achieved by the former are both higher than those of the latter. That is because the former is acquired by combining descriptors in ChemoPy and RDKit databases, including one-dimensional information in addition to two-dimensional information, compared with the latter. On the other hand, it is possible for the Mordred descriptor set that there is redundant information among descriptors, which influences extraction of structure information so as to degrade classification performance. In a word, comparing with other descriptor sets, combinatorial descriptor set is more appropriate to extract information from the collected drug molecular structure to obtain better classification performance. The detailed computation procedure for five descriptor sets is presented in Section 3.1.2.

#### 2.1.3. The Impact of Classification Models

According to results in Table A2, Table A3, Table A4, Table A5 and Table A6 of the Appendix B, the fusion method obtained by RF and SVM performs better than single classifiers and other fusion models. The implementation principles and types of fusion methods are introduced in Section 3.2.2. The classification results by five single classifiers and seven fusion methods based on combinatorial descriptor data from molecular set S_4_ are displayed in Figure 4. The Q value and kappa coefficient are represented by different colored bars in the graph. Comparing all results by single classifiers, it can be found that better classification performance is achieved through SVM. In particular, classification performance obtained by LDA is far poorer than that of other classifiers, indicating that the correlation is not a simple linear relationship. Comparing carefully all fusion results, it can be observed that the DS12 method, fusing RF and SVM, performs better than other fusion methods and outperforms individual SVM. However, there are four fusion methods whose results are not better than that of single SVM, such as the fusion method based on SVM and LR, by which the obtained Q value and kappa coefficient are both lower than SVM by nearly 0.01.

Regarding fusion method, theoretically, the greater the number of fusion classifiers, the better the classification results. In fact, the result obtained by fusion method is limited by the performance of each classifier and could be worse than that of all classifiers due to evidence conflict. The result achieved by DS12345, fusing five classifiers, is far poorer than that of DS1235 fusing four classifiers, owing to the poor performance of LDA. Comparing results of DS12, DS123, and DS1235, due to the poor performance of LR, result obtained by DS123 is worse than that of DS12, but the result obtained by DS1235 is not as bad as that of DS123, although the performance of ABT is even poorer. As mentioned in Reference [27], the relationship between individual classifiers has not been considered in fusion method, while in Reference [28], only great conflicts between single classifiers are processed according to an improved method. This may be explained by the fact that there are conflicts between the result of LR and the two results of RF and SVM, while there is no conflict between the result of ABT and the three results of LR, RF, and SVM. Additionally, concerning certain cases where LR performs better than RF, as shown in Table A3 of the Appendix B, the results still show that the method fusing RF and SVM outperforms that of LR and SVM. This indicates that there is conflict between the result by LR and the result by SVM. Moreover, it demonstrates that RF is applicable to deal with uneven data sets, which makes results of RF and SVM complement each other to achieve better classification performance.

Based on the discussion in Section 2.1, the highest Q and K are obtained based on combinatorial descriptors, which are 0.862 and 0.81, respectively. The obtained standard deviation of kappa index is also the smallest, 0.028, as shown in Table A2 of the Appendix B. The result illustrates the applicability and reliability of combinatorial descriptors in predicting unknown compounds. Although a reliable prediction model has been obtained by KS method, the imbalance in molecular sets does have an influence, so the classification results based on different molecular sets cannot be more objectively measured and compared. Hence, according to the obtained results, the prediction performance of single SVM and two fusion models are further verified by the external validation set.

### 2.2. The Analysis on External Validation Set

In order to evaluate models more objectively, the diversity of the external validation set should be ensured. Aiming at different categories of drug data, partial random data are added into the external validation data set from S_1_, S_2_, S_3_, and S_4_. Seven classes of single-role drugs, whose order is consistent with that of the collected molecular sets, and all multi-role drugs are also contained in the final validation set, and their amounts are 7, 21, 26, 10, 5, 6, 2, and 10, respectively. The statistical mean obtained by running the procedure three times is used as the prediction result for the external validation set. Based on five descriptor data from four molecular sets, prediction results for external validation molecules are shown in Table 1, which are all obtained by four classification methods that perform better in Section 2.1.3. The prediction for the validation molecules is shown in Appendix B Table A7.

In terms of combinatorial descriptors, better prediction for validation molecules is achieved with molecular set S_3_. For Mordred descriptor data, good classification for the external validation set is implemented in molecular set S_2_. With regard to MACCS fingerprints, the highest correct prediction number is obtained with molecular sets S_2_ and S_3_. For topological fingerprints, the better prediction result is achieved with molecular sets S_1_ and S_3_. When Morgan fingerprints are used as descriptor data, better classification results are obtained with molecular set S_2_. From these results, it can be found that the greatest correct prediction number is achieved based on combinatorial descriptors. Moreover, the result obtained based on descriptor data from molecular set S_3_ performs well in most cases, and the prediction achieved by topological fingerprint data only performs well from S_1_.

To further conduct comparative analysis, better results, obtained by different descriptor data and molecular sets, are displayed in Table 2. For both Mordred descriptor set and topological fingerprint set, the results from molecular set S_3_ are listed in Table 2. Four descriptor data calculated from molecular set S_4_ are all trained by DS12 method to predict molecules in the validation set. The detailed prediction results for each class are shown in Table 2. The correct prediction number of single-role drugs for different classes and the correct prediction number of multi-role drugs are also included. By using combinatorial descriptor data from molecular set S_3_ to train DS12 method, the highest correct prediction number is obtained. However, it can be found that the correct prediction number of DS12 for antiarrhythmics is zero, indicating that prediction performance of DS12 is not balanced. Comparing carefully the correct prediction rate of each class of single-role drugs, most results are acceptable. From the aspect of whole prediction performance, four models with better performance are bolded in the first column of Table 2. Based on them, two multi-role drug molecules are taken as examples to further verify the four models, whose prediction probabilities for each class are listed in Table 3.

These two drug molecules are rifampicin and celecoxib. Rifampicin, as an antibiotic, has antitumor activity [29]. It is consistently correctly predicted as antibiotic by the four models. Meanwhile, prediction for its second possible activity is also consistent with reality for all four models. Celecoxib is used as an analgesic at first and then is used as an antitumor agent because of its favorable antineoplastic properties. Furthermore, studies have verified antiviral efficacy of celecoxib since 2013 [30,31,32]. From the prediction probabilities in Table 3, it can be found that correct prediction for analgesic and antineoplastic activities of celecoxib cannot be obtained by a single SVM. Prediction only for antiviral efficacy of celecoxib is incorrect for these two models based on topological fingerprints. It is worth noting that celecoxib is considered more likely to be an antidiabetic drug, based on the result of combinatorial descriptor set, instead of an antitumor and antiviral drug, which needs to be further confirmed.

In summary, drug information included in molecular sets S_4_ and S_3_ is more helpful to establish correlation between drug molecular structure and clinical therapeutic effect, which provides more reliable prediction for unknown drugs. The combinatorial descriptor and topological fingerprint are favorable for extracting structure information, which facilitates the mining of the correlation. Furthermore, compared with single classifier, higher Q value and kappa coefficient are obtained by fusion method, which is more suitable for predicting the potential clinical therapeutic effect of unknown drugs.

## 3. Materials and Methods

Analgesics, antineoplastic drugs, antibacterial drugs, antiviral drugs, antifungals, antidiabetic drugs, and anti-arrhythmic drugs are taken as examples to conduct multi-classification research on drugs. Section 3.1 details the collection of drug molecules and the acquisition of drug data. The study procedure of classification, the basic theory, and the fusion method that classification models depend on are introduced in Section 3.2.

### 3.1. Drug Collection and Corresponding Descriptor Data Set

After collecting drug molecular information, drug molecular structure is converted into a form that can be recognized by computer by Python script PubChemPy (https://pypi.python.org/pypi/PubChemPy, accessed on 10 March 2022). Based on this form of structure, five descriptor sets are calculated by embedding ChemoPy [33], Mordred [34], and RDKit [35] packages into Python.

As mentioned above, all drugs can be considered as certain molecules or a collection of molecules with certain structures. There are several available databases including known drug information, such as commercial names, molecules, basic physical-chemical properties, and structure descriptors, simplified molecular linear specifications (SMILES). Depending on the area and focus of their developer, these databases may cover different drugs and corresponding attributes, which are usually presented in different data formats. To include drugs that are from more categories and described by more universal attributes, drugs from three databases are collected and grouped into seven classes according to their actual clinical therapeutic effects.

Many properties are quantitative expressions of molecular structural information, which can be calculated by software or even just web applications [30,31,32]. Here, five different types of description data are chosen to select the best one. The SMILES is often used as input form for computational programs to calculate the descriptor data, which include general information on molecular structure. Drug data analyzed in this work are detailed in the following section.

#### 3.1.1. Drug Molecules

Drug molecules in seven categories are initially collected from World Health Organization official website and then are checked according to the drug information in KEGG [8], DrugBank [9], and PubChem [36]. During the collection of drugs, all compounds with therapeutic effects were included, such as prodrugs and active metabolites. The comprehensive coverage of drug molecular sets is ensured, so that more reliable prediction ability is obtained. According to actual clinical therapeutic information, drug molecules belonging to one class may be classified as another category or removed from a drug molecular set. It can be found that drugs with the same molecular structure in different databases may be named differently. In this case, the repeated ones are removed from the database.

#### 3.1.2. Different Descriptors

Molecular digital representation is varied, including both experimental and computational properties. The computer-acquired descriptive properties are widely applied due to their convenience and usability, such as molecular descriptors and molecular fingerprints [37,38], and they are used to facilitate drug development [13,16]. Although there are still many types of data for quantifying molecular structure, such as three-dimensional descriptors and pharmacophore fingerprints, there are limitations to them, such as high computational complexity and slow computational speed, which have a key impact on drug screening. In order to achieve better representation of drug molecular structure as soon as possible, five data sets quantifying molecular structure information from various aspects are calculated and used as feature data to classify drugs to determine better description for drug structure. They can be calculated by programming software, as detailed in the following.

Two types of molecular descriptor sets, whose data are real numbers, are selected as description data for drug molecules. Various descriptor groups are formed based on multiple descriptors obtained by different calculation methods. The computation of multiple descriptor groups has been implemented, and the relationships among them also have been clearly shown [39]. Here, the combination of ChemoPy [33] and RDKit [35] descriptors is chosen as a descriptor set. After removing repeated descriptors, the combinatorial descriptor set can be obtained from these two databases. It contains 632 descriptors from ChemoPy and 123 descriptors from RDKit. In addition, a Mordred descriptor group proposed by Moriwaki et al. [34], including multiple sets of descriptors, is used as another descriptor set. To ensure data is processable, a total of 1127 Mordred descriptors are reserved due to the missing value of other descriptors. Two descriptor sets are calculated by embedding ChemoPy, Mordred, and RDKit packages into Python.

Data for molecular fingerprints are all binary. Molecular fingerprints are diverse [40,41], and each fingerprint is a binary vector with certain dimensions. There are three types of fingerprints with fixed dimensions. Morgan fingerprints, MACCS fingerprints, and topological fingerprints are selected as another three descriptor sets to acquire data for drug molecules. Especially for Morgan fingerprints, when the selected circle radius is different, the vector dimension of the fingerprint is also different. Here, the radius is set to 4, and a 1024-dimensional Morgan fingerprints is obtained. They are all generated by combining RDKit database and Python program.

#### 3.1.3. Final Molecular Set

When calculating the above descriptor data, it is found that partial molecular structure information cannot be converted into complete and processable data, such as mixtures, ionic compounds, and biological macromolecules. Molecules with molar mass more than 1800 are unsuitable to train models with other drug molecules, because their values are much higher than others. Drug molecules that match the above conditions are removed.

Although five types of descriptor sets cannot distinguish isomers, they have been widely used in drug screening [13,14,15,16]. Furthermore, they do not perform any worse than the data for describing three-dimensional structure [42], whose computation procedure are complex and computation time are long. Therefore, it is suitable to classify drugs through the calculated data, and isomer pairs are removed during the checking phase of drug molecule collection.

Additionally, some drugs for the treatment of certain diseases also have other therapeutic uses, as shown in References [43,44]. To obtain more explicit classification results, only molecules with a single therapeutic purpose are kept in the analysis data set, and drugs with multiple therapeutic effects are collected into an external validation set. After removing some drugs in the early stage, an original molecular set S_1_ containing 1019 drug molecules is obtained.

In order to collect as many drug molecules as possible with a certain therapeutic effect, namely analgesic effect, analgesics and several anesthetics and antipsychotics that achieve analgesic effects only because of local anesthesia or muscle relaxation are all contained in molecular set S_1_. When carefully checking therapeutic use and mechanism, it was found that several drugs are just healthcare products or adjuvants, such as radiation adjuvants. Therapeutic effects of certain drugs also cannot be confirmed according to articles on PubMed website [45]. Additionally, a molecular set that is appropriate to be used as basic data set for identifying unknown drugs is currently unable to be determined. It is necessary to check drug-relevant information and remove some drug molecules to obtain a molecular set that is more beneficial to discovering the relationship between structure and properties. Database information and literature information from PubMed are collected and checked, including ATC code, research phase, target, therapeutic effect, mechanism of action, and applicable subjects. The original data set is screened layer by layer according to concrete characteristics of drug molecules, and then four molecular sets containing different numbers of drugs are obtained. The detailed process of acquiring four different molecular sets is as follows.

After removing 19 drugs with weak relief for pain and sedative effect or auxiliary, 2 antineoplastic drugs with only healthcare effect and inhibiting DNA repair, and 2 antiviral drugs in S_1_, the molecular set S_2_ containing 996 drugs is obtained. Based on S_2_, drugs with potential therapeutic uses are removed, and then the molecular set S_3_ containing 921 drugs is obtained. Based on S_3_, drugs without known mechanism, still used as veterinary drugs or just in the clinical trial phase are removed, and thus molecular set S_4_ with 844 drugs is obtained. The acquisition process of the four molecular sets is illustrated in Figure 5. While the seven types of drugs are collected, drug molecules with two or even three therapeutic uses are identified and stored. Number of drugs included in four molecular sets is summarized in Table 4. Additionally, an external validation molecular set containing 37 drugs is obtained. For all collected molecules, their name and category can be seen in Appendix A.

### 3.2. Methods for Selection, Combination, and Evaluation

The prediction model is established based on classification method. The performance of the final model depends on the applied classifier and its parameter setting. For constructing models with better classification performance, multiple classifiers are applied to obtain classification models based on DS fusion method. For objective comparison and selection, suitable methods and indicators should be used to evaluate different models. Digitalized descriptors are generated for further machine learning algorithms.

The whole classification procedure is accomplished with Python. Five different types of classifiers are utilized to extract drug information by Python program and scikit-learn packages [46]. They have been described in detail in Reference [47].

#### 3.2.1. Classification Algorithms

In order to achieve multi-classification and ensure interpretability of results, strategy for multi-classification based on binary classifiers is applied. There are two ways to proceed according to the strategy, namely “one vs. rest” (“OvR”) and “one vs. one” (“OvO”). Considering the computational cost and subsequent fusion processing issues, “OvR” is adopted here. Five algorithms are introduced as follows.

RF had been widely used in classification since the beginning of the 21st century. The number of decision trees for RF, as a key parameter, has an key impact on the performance of the algorithm. In order to increase the speed of optimization and computation, the adjustment range of the parameter is set from 10 to 100 and incremented by 10. As another parameter that needs to be adjusted, the split criterion can be set as Gini or entropy.

ABT, similar to RF, is also a tree-based classifier. Different from RF, which is conducted by directly adding and averaging the results of a large number of trees, ABT is trained by gradually emphasizing the weight of those samples that are difficult to separate based on the classification of each tree. The changing range of basic classifier number is the same as that in RF.

SVM is a classifier defined in feature space, which had been widely used in statistical classification and regression analysis. A hyperplane was constructed by SVM to separate the training samples. It was employed to process linear and non-linear data through the kernel function, and its classification accuracy is closely related to the kernel function. The penalty parameter C is an important parameter, and its range is 1 to 10. After previous attempts, the polynomial and the radial basis function kernel are more suitable for the obtained data set. In grid search, the kernel function varies between the above two kernels.

As a standard two-classification method, LR was adopted to realize classification by similar regression, i.e., to calculate the relationship between the conditional probability of each sample feature vector and the set threshold to determine the sample class. Its performance depends on whether the data conform to the predetermined model. Since “OvR” strategy is applied to achieve multi-classification, “liblinear” is chosen as the solver according to the preliminary attempts. As the key parameter, the value of penalty factor C varies between 0.0001, 0.001, 0.01, 0.1, 1, and 10.

LDA was an extension of Fisher discriminant analysis. It was a weighted linear combination between a set of feature variables. This set was obtained by training data under the condition that the variance within classes was as small as possible and the variance between classes was as large as possible. It is used as a decision function to recognize the category to which the sample belongs. Default parameters for LDA are adopted in this work [46].

#### 3.2.2. Fusion Methods

After the model framework is established, the best classification model should be selected by validation methods. The division of data sets affects the prediction results of the model for the test set and will eventually affect the choice of the best model. As is well known, a single classifier is usually suitable to a certain scope. Multiple classifiers are applied to the obtained data set. Moreover, the collected data set is multi-category and unbalanced, and it is more difficult to obtain proper division for the data set. To achieve better classification performance, many fusion methods have been proposed. Classifier fusion methods are different in architectures and ways of fusion, and the DS fusion method is applied in this study. The DS fusion method was introduced and developed in the work of Dempster and Shafer [48,49]. This theory has been applied in many fields, such as fault diagnosis, and there are also examples in the field of classification [27,28].

There are multiple situations for the categories of collected sample data, which are represented by a limited nonempty set Θ. Enumerating the possible categories of data sample, $\Theta =\left\{\theta_{1},\theta_{2},\cdots ,\theta_{c}\;\right\}$ can be obtained, where c represents the number of hypotheses. In DS theory, Θ is called the discrimination frame, and $2^{\Theta }$ represents a power set containing $2^{c}$ cases, namely $2^{\Theta }=\left\{\phi ,\left\{\theta_{1}\right\},\cdots ,\left\{\theta_{c}\right\},\left\{\theta_{1},\theta_{2}\right\},\cdots ,\Theta \right\}$. Corresponding mass function m can be obtained for each case in the set Θ, which is also called basic probability assignment (BPA) ranging from 0 to 1. Under this assumption, BPA should meet the following two conditions:(1)$$m\left(\emptyset \right)=0$$
(2)$$\underset{A\in 2^{\Theta }}{\sum }m\left(A\right)=1$$
where $m\left(A\right)$ is the BPA of a certain situation and is also the confidence of its occurrence. Therefore, when BPA is 0, the confidence of a certain situation is 0, and the opposite is 1. According to Dempster’s combination rule in [39], the confidence level (in some case A) obtained by fusion of two pieces of evidence is calculated as follows:(3)$$m\left(A\right)=\left\{\begin{array}{c} 0\;\;\;\;\;\;\;\;\;\;\;\;\;\;\;\;\;\;\;\;\;\;\;\;\;\;A=\emptyset \\ \frac{1}{1-K_{c}}\underset{B\cap C=A}{\sum }m_{1}\left(B\right)m_{2}\left(C\right)\;\;\;A\neq \emptyset \end{array}\right.$$
where $K_{c}$ represents the conflict between the confidence of the two evidences. The calculation formula is in Equation (4):(4)$$K_{c}=\underset{B\cap C=\emptyset }{\sum }m_{1}\left(B\right)m_{2}\left(C\right)$$
where *B* and *C* indicate possible situations under the evidence system.

When the number of evidences exceeds 2, the fusion is achieved according to Dempster’s combination rule, and the calculation rule is as follows:(5)$$m=m_{1}⨁\;m_{2}⨁\cdots ⨁m_{n}=\left(\left(\left(m_{1}⨁m_{2}\right)⨁\cdots \right)⨁m_{n}\right)$$
where ⨁ represents an operation that can fuse two classification results. As explained in References [27,28], DS evidence fusion is not applicable when there are conflicts between evidences. Since the $K_{c}$ value between the evidences reflects the degree of conflict between the evidences, for the case where $K_{c}$ is too large, DS fusion should be replaced by other methods. In Reference [28], a different fusion method is proposed afterwards for the case of $K_{c}$ > 0.95. For this situation, the method mentioned in Reference [28] is also applied in this paper.

All fusion models applied in Section 3 are shown in Table A1 of the Appendix B.

#### 3.2.3. The Evaluation of Classification Performance

An appropriate model evaluation method is the key to discovering models with very good classification performance. It compares performance of different methods based on test sets obtained by dividing data sets. There are many methods for evaluating models now, such as common leave-one-out and bootstrapping. However, these methods are more suitable for small and balanced data sets, which are inconsistent with the characteristics of the collected data sets. In terms of the data set, leave-one-out has high computing cost, and bootstrapping does not make full use of data information.

To get objective statistical results under the influence of a data set in random sampling, the KS method was proposed first by Kennard and Stone [26] and applied and compared in research by Martin et al. [50]. The KS method is selected as the division method of data sets in this study. The training set and the test set are obtained based on difference between samples, so that the model obtained by means of the test set with diverse samples can achieve a more reasonable and reliable classification prediction for unknown drugs.

Because the difference is measured by distance between samples in KS method, the data need to first be standardized. Based on normalized training set, key parameters for each classifier are determined by grid search method that is implemented by five-fold cross-validation. Then, molecules in the test set are classified by classification methods with the optimized parameters. For comparability of methods and objectivity of results, KS division is repeated 100 times to obtain statistical results. Moreover, the model for better and more robust classification can be obtained by comparing the results of different classification methods.

In addition, it is also important for model evaluation to select appropriate evaluation indicators, which will determine the reliability of a model. As one of the evaluation indicators for classification models, prediction accuracy (*Q*) is most commonly used and most intuitive. It can be calculated by Equation (6).
(6)$$Q=\frac{the\;number\;of\;sample\;predicted\;correctly}{the\;total\;number\;of\;sample}$$

Compared with various evaluation indicators of two-class classification, there are fewer indicators that can be used for multi-class. There is currently a metric, Cohen’s kappa coefficient (*K*), which has been widely used to evaluate performance of two-class and multi-class models. *K* is calculated as follows:(7)$$K=\frac{Q-p_{e}}{1-p_{e}}$$
where $p_{e}$ is quotient obtained by dividing the sum of the products for each class by the square of the number of samples, and the product refers to the number of samples that actually belong to a class times the number of samples predicted to be in the class. The range of *K* is from −1 to 1. The larger its value, the more consistent the predicted result is with the real situation.

#### 3.2.4. Study Process of Classifying Drugs

The overall flow of this study is shown in Figure 6. Different data sets compared in Figure 6a come from the collocation and combination of different molecular sets and different descriptor sets in the study. The different types of molecular sets and descriptor sets and all applied classification models can be found in Table A1 of the Appendix B, and the details of the classification process are as follows.

An example for model selection is obtained by randomly choosing a molecular set and a descriptor set from the first two columns of Table A1. Their combination with all the classification models in the third column of Table A1 is used to implement classification procedures as shown in Figure 6a. The molecular set is input for calculating the descriptors set. The calculated descriptor set is grouped into training set and test set by KS method. The training set is used to tune parameters of the classification algorithms and fit classification models. Then, the test set is classified by fitted models, and their results are evaluated and saved. This process, from grouping to evaluation, is run 100 times to obtain a statistical average result from different models. By comparing classification indicators, the better model based on chosen molecular set and descriptor set will be discovered. In this way, the better models for each combination of molecular set and descriptor set are selected to identify the molecules of the external validation set.

The whole prediction procedure for external validation is displayed in Figure 6b. Similarly, an example for external validation is obtained by randomly choosing a molecular set and a descriptor set in Table A1. Classification models are selected from those that perform well in Figure 6a. The chosen molecular set and molecular set for external validation are used as the input for calculating the descriptor set at the same time. In order to ensure the class diversity of final validation molecules, descriptor data of partial molecules are randomly selected for the external validation descriptor set from the descriptor set calculated based on chosen molecular set. Afterwards, the rest of the descriptor set is utilized to tune parameters of the classification algorithm and fit the classification models. The renewed external validation descriptor set is predicted by the fitted model, and the prediction probabilities of different models for it are saved. Differing from the study flow in (a), this procedure is run 3 times to obtain a final statistical predicted probability result.

## 4. Conclusions

Compared with the traditional virtual screening of a single class of drugs, virtual screening based on multiple classes of drugs not only enhances the screening efficiency but also discovers multiple possible uses of a drug at one time. In this study, seven classes of drugs are taken as examples to obtain enough drug structure information from various databases. Structural information on drug molecules can be converted into feature information such as descriptors and fingerprints based on SMILES. Then, a DS fusion model based on five classifiers is proposed to make full use of descriptor data for good prediction performance. Subsequently, by comparing the classification results, better methods for classifying multi-class drugs are found, including the drug structure description method and the machine learning method. Based on the above results and discussions, it is found that combinatorial descriptor data is more appropriate to extract drug information to obtain better classification performance. Compared to the single classification methods, SVM performed better in multi-class drugs classification, indicating that there is a nonlinear correlation between drug molecular structure and treatment effect. Additionally, the established fusion methods outperform the single machine learning methods, especially the DS12 method fusing RF and SVM. The final results suggest that the combination descriptor data and DS12 classification method can make a better prediction for multi-class drugs, which is also verified by the results from the external verification set. This study provides a methodological basis for simultaneous screening of multi-class drugs and a new direction for speeding up virtual screening.

Although a good classification result is obtained, the study is only focused on discovering correlation between a drug’s therapeutic effect and its two-dimensional structure, ignoring the effect of drug isomers on therapeutic effects, which needs further research. In addition, the current classification of drug therapeutic uses is rough. In fact, there are far more than seven classes of real drugs, and each class can be further subdivided. These problems can be gradually explored and solved through in-depth research, such as adding a non-drug class and further expanding the drug classes.

## Figures and Tables

**Figure 1 molecules-27-04807-f001:**
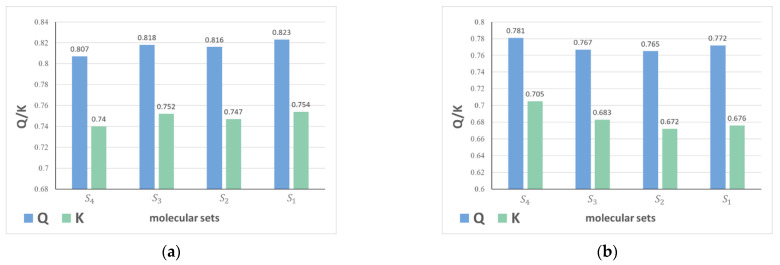
Classification results obtained by RF. (**a**) Results from Mordred descriptor set; (**b**) Results from Morgan fingerprint set.

**Figure 2 molecules-27-04807-f002:**
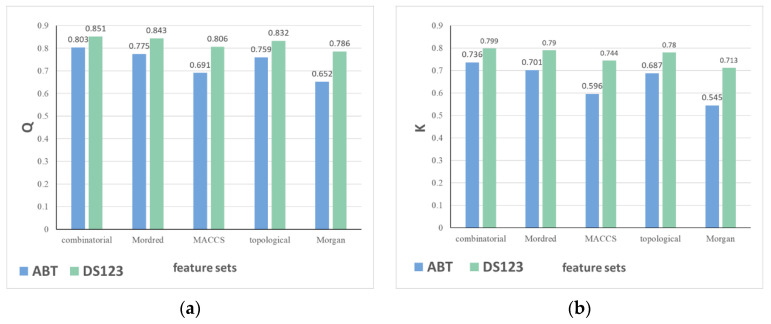
Classification results based on molecular set S_4_. (**a**) Q values; (**b**) Kappa coefficient.

**Figure 3 molecules-27-04807-f003:**
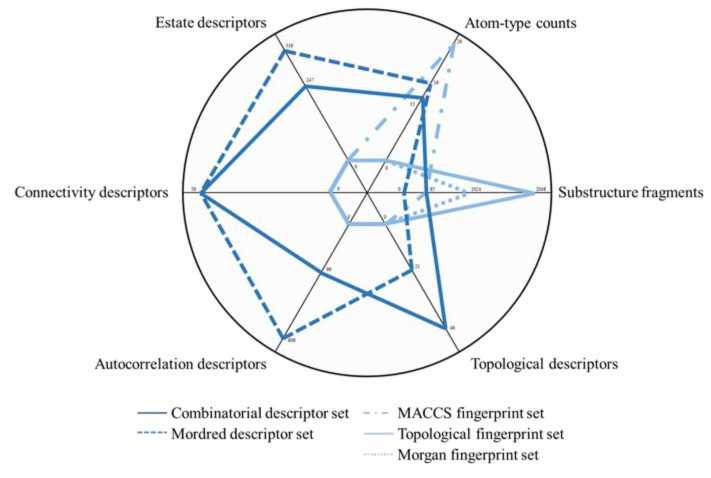
Six types of descriptor information included in five data sets.

**Figure 4 molecules-27-04807-f004:**
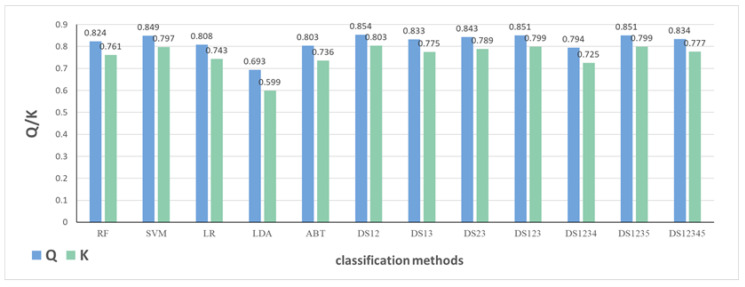
Classification results based on molecular set S_4_ and combinatorial descriptors.

**Figure 5 molecules-27-04807-f005:**
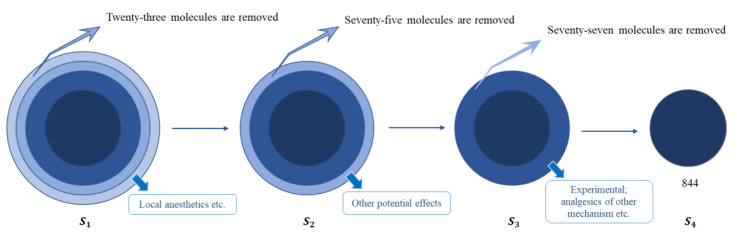
The acquisition of molecular sets. S_2_ is obtained by checking drug therapeutic mechanism. S_3_ is obtained by checking other potential therapeutic effects. S_4_ is obtained by checking applicable objects and experimental stage.

**Figure 6 molecules-27-04807-f006:**
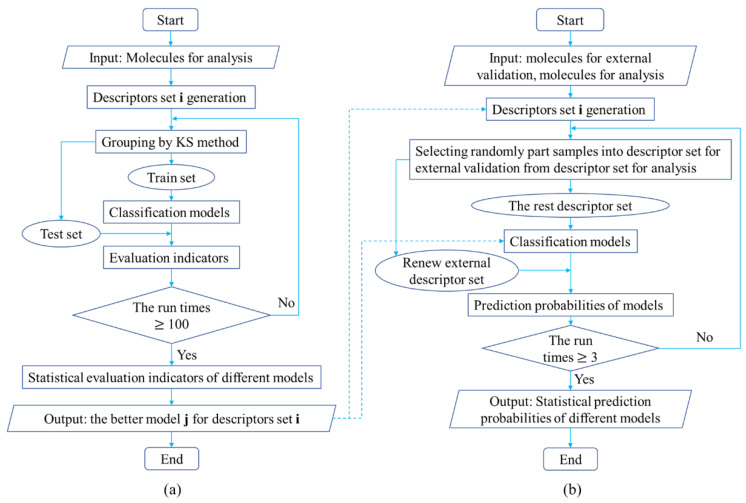
The whole study process, where (**a**) is the flow for comparing classification results based on different data sets and (**b**) is the flow for further validation.

**Table 1 molecules-27-04807-t001:** The number of correct predictions by fusion method in the external validation set.

Descriptor Sets	S_4_	S_3_	S_2_	S_1_
Combinatorial descriptor	73	76	73	70
Mordred descriptor	74	74	75	72
MACCS fingerprint	70	74	74	71
Topological fingerprint	74	75	74	75
Morgan fingerprint	73	69	75	70

**Table 2 molecules-27-04807-t002:** The correct prediction for each class of drugs by several models. In first column, F_1_–F_5_ are different descriptor data, and detailed information is shown in Table A1 of the Appendix B. The seven classes of drugs are represented by C_1_, C_2_, C_3_, C_4_, C_5_, C_6_, and C_7_ in order. The same is true for Table 4.

Models	Single-Role								Multi-role/10	Total/87
	C_1_/7	C_2_/21	C_3_/26	C_4_/10	C_5_/5	C_6_/6	C_7_/2	Total/77		
**F_1_—DS12 on S_3_**	4	18	26	8	5	6	0	67	9	76
F_1_—DS12 on S_4_	4	16	25	8	4	5	2	64	7	71
**F_2_—SVM on S_2_**	4	16	26	6	5	6	2	65	10	75
F_2_—DS12 on S_3_	4	17	26	7	4	3	1	62	10	72
F_2_—DS12 on S_4_	4	14	25	8	5	5	2	63	10	73
F_3_—SVM on S_3_	4	15	26	7	5	5	2	64	10	74
F_3_—DS12 on S_4_	3	15	26	5	5	4	2	60	8	68
**F_4_—DS123 on S_3_**	5	16	25	6	5	6	2	65	10	75
**F_4_—DS12 on S_3_**	5	17	25	5	5	5	2	64	10	74
F_4_—DS12 on S_4_	4	16	26	5	5	5	2	63	10	73
F_5_—SVM on S_2_	6	13	26	4	3	4	0	56	9	65
F_5_—DS12 on S_4_	5	15	26	7	3	6	2	64	9	73

**Table 3 molecules-27-04807-t003:** The predicted probabilities for the four drugs.

Drugs	Models	Classes						
		C_1_	C_2_	C_3_	C_4_	C_5_	C_6_	C_7_
Rifampicin	F_1_—DS12 on S_3_	0	0.005	0.994	0	0	0	0
	F_2_—SVM on S_2_	0.028	0.079	0.833	0.012	0.023	0.008	0.017
	F_4_—DS123 on S_3_	0.002	0.005	0.991	0.001	0	0	0
	F_4_—DS12 on S_3_	0	0.001	0.998	0	0	0	0
Celecoxib	F_1_—DS12 on S_3_	0.395	0.1	0.003	0.046	0.008	0.447	0
	F_2_—SVM on S_2_	0.387	0.08	0.15	0.117	0.025	0.236	0.005
	F_4_—DS123 on S_3_	0.535	0.321	0.053	0.038	0.009	0.041	0.002
	F_4_—DS12 on S_3_	0.609	0.318	0.021	0.012	0.004	0.035	0.001

**Table 4 molecules-27-04807-t004:** Drug molecules included in four molecular sets.

Drug Classes	Molecular Set S_1_	Molecular Set S_2_	Molecular Set S_3_	Molecular Set S_4_
Analgesics	228	209	183	164
Antineoplastic	211	209	189	165
Antibacterial drugs	296	294	285	261
Antiviral drugs	108	108	102	99
Antifungals	64	64	57	54
Antidiabetic drugs	70	70	66	63
Antiarrhythmics	42	42	39	38
Total	1019	996	921	844

## Data Availability

All drug molecules utilized in this work are acquired from KEGG database (https://www.kegg.jp/, accessed on 5 January 2022), DrugBank database (https://go.drugbank.com/, accessed on 15 April 2022), and PubChem database (https://pubchem.ncbi.nlm.nih.gov/, accessed on 15 April 2022).

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
