# Peer review of "Small Molecular Drug Screening Based on Clinical Therapeutic Effect"

_molecules, 2022, doi:10.3390/molecules27154807_

Round 1
Reviewer 1 Report
The set of marketed drugs is a very biased collection of very privileged molecules, and it should, in my opinion, not be used as such for machine learning - because it does not offer enough structure-activity information, as it mostly consists of "singletons" or small families of very closely related compounds. For example, benzodiazepines is a typical family of drugs - they all have the benzodiazepine scaffold in common. Thus, it is very easy for a machine learning tool to basically "learn by heart" these key families - benzodiazepines, kinase inhibitors (all ATP-like), steroid (self-explanatory), macrocycles (antibiotics) etc. The problem with your approach is that you try - and succeed - to teach your model to discriminate between benzodiazepines and antibiotics, and this is very simple to achieve. However, your model will ALSO predict that ALL more or less close analogues of the benzodiazepine drug as "benzodiazepine" and ALL analogues of your macrocyclic ring as "antibiotics". This is wrong - most macrocycles are NOT useful as antibiotic drugs and most compounds that can be synthetized around a benzodiazepine scaffold are NOT benzodiazepine receptor inhibtors! But, in order to do this, you'd need a training set including not only the examples of successful drugs, but also all the unsuccessful analogues that had to be discarded on the way to the successful drug! Otherwise, your model is pretty much useless - in the sense that it would not do more than "rediscovering" the wisdom of any student of pharmacy - detecting the main therapeutic classes.
And... the English must be reviewed by some native speaker because... occasionally it is outrigh hilarious! "Li et al. proposed a support vector machine model to identify drugs with liver injury"!! I am sorry, I did not know that some drugs have livers, and some of them suffer of liver injury! They should be immediately provided medical care, rather than exploited for machine learning in spite of their obvious distress!
Author Response
Dear Ms. Katarina Modic and reviewer,
Thank you very much for taking your time to review this manuscript. We appreciate your comments and suggestions. Please find the itemized responses in the attachment and our revisions in the resubmitted files.
Looking forward to hearing from you.
Many thanks and best regards,
Sun Wei

Reviewer 2 Report
An acceptable revised version was presented along with a detailed rebuttal letter following the recommendations and comments.
Author Response

(The authors gave the same response as above.)

Reviewer 3 Report
Comments on the manuscript ID molecules- 1811241 by Zhong et al. titled “Small Molecular Drug Screening Based on Clinical Therapeutic Effect”:
The study is focused on discovering correlation drug therapeutic effect/drug two-dimensional structure. A multi-classification method based on Dempster-Shafer evidence reasoning theory is proposed to predict possible clinical therapeutic effect of unknown drugs. A relatively good prediction for potential clinical therapeutic effect of unknown drugs is achieved based on the collected set of 1019 drug molecules. The manuscript has shortcomings in the general writing and in the presentation and discussion of the results.
Minor points:
· Introduction, p.1, lines 31-32: “State Food and Drug Administration” is the former China's medical device market regulator, each country has its own regulator;
· Introduction, p.1, line 44: Dock – docking;
· Introduction, p.2, line 74: “drugs can be categorized into different clinical therapeutic effects” – “the drugs can be categorized in different categories…” or “the drugs can be classified by what they treat”;
· p.3, line 110: “are selected as the single classifier”;
· p.3, line 142: “For each table, its result is obtained”;
· Results and Discussion, p.3, lines 127-132: this text should be removed, it summarizes undiscussed results;
· p.3, lines 134-140: again, some molecular sets are called “more suitable”, “better”, etc. before presenting results;
· p.4, line 158: “by checking drug information that is from many aspects” – many-sided, multifaceted information?
· Figure 1: why the data sets are in reverse order (from S4 to S1)?
· p.5, lines 200-201: “computation procedure of” – “computation procedure for”
· p.6., line 240: “Based on the discussion in section 3.1” – do not discuss results given below in the text;
· Table 1 – reverse order of Ss;
· p.15, line 600: “Although better classification result is obtained” – better than?
Conclusion/recommendation: The manuscript requires some improvement. I recommend a minor revision before it could be considered for publication.
Author Response

(The authors gave the same response as above.)

Round 2
Reviewer 1 Report
It seems that my previous comments were a bit lost on the authors - yes, of course, we all use molecular descriptors, not brute structures - my previous comments were of course referring to the current state-of-art in chemoinformatics. The fact that a dimethyl-amine containing drug (!) was correctly predicted does not mean anything: Me2N is a functional group appearing in many drug classes, and the proper one in your case was detected based on the greater structural pattern/scaffold. I did not cite Me2N in my comments, but larger structural patterns - steroids, benzodiazepines, ATP-like compounds, peptide-like compounds, bioactive amines, etc. Note that most of Me2N containing molecules (there must be some 10 to the power of 20 of those) are not and never will be drugs at all - something your method is not trained to recognize, because you only train on drugs! There is no point to insist - if you don't enhance your training sets with non-drugs, your models will be very biased and therefore useless.
Author Response
Thank reviewer for the concern of this issue. As reviewer said, our model only obtains information about the seven classes of drugs that we collected, and the compounds whose categories are not included in the training set are not considered. It may be classified incorrectly when a compound does not belong to any of these seven classes of drugs at all.
The idea reviewer mentioned is truly a sound research plan, which would facilitate drug development a lot. However, there is a slight difference between the plan proposed by reviewer and the one proposed by us. The focus of this study is not to screen out these seven classes of drugs from a large number of compounds, but to use these seven classes of drugs as examples to explore descriptor data and machine learning methods that are more conducive to classifying multiclass drugs. The significance of this work is to provide a reference for virtual screening of multiclass drugs simultaneously.
The existing virtual screening mostly focuses on screening drugs with single therapeutic effect. If there is a drug with multiple therapeutic, the computational complexity of information extraction will gradually increase. It may be quite time consuming to train the models. At the same time, screening multiclass drugs can know all these classes information at one time, which can directly predict all possibilities of one drug's therapeutic. In this way, drugs with multiple therapeutic can be detected earlier and faster. At present, there are many kinds of classification methods of drugs from therapeutic effect and molecular structure. Different drug structure description methods and machine learning methods will have different impacts on the classification accuracy. There is few investigation on which description method can more effectively describe the structural information of drugs and which machine learning method is more conducive to multi-classification of drugs.
Therefore, the purpose of this study is to provide a basis for multi-classification of drugs, including finding more appropriate drug description methods as well as machine learning methods. The drugs used in the training set, test set, and external validation set are all collected from the data platforms (KEGG, DrugBank and PubChem) where the effect of drugs is confirmed, and the effect of the drug that belongs to these seven categories is double check by consulting literatures. In this way, although there are biases in the current model as reviewer said, the conclusion of this paper is not affected, that is, which descriptor data and machine learning methods can more accurately classify drugs based on the correlation between drug structure and effect. Thus, there is no case of misclassifying a compound that does not belong to these seven classes of drugs into one of these seven classes. Furthermore, the kappa coefficient of most of the methods in this paper is above 0.61, and the highest can be 0.81, which meets the basic requirements of accurate classification (The kappa coefficient greater than 0.61 indicates that the predicted result is close to the actual situation). These results also reflect the reliability of the research conclusion from the other side (see Table A2 of appendix). The external verification results also proved the effectiveness of the research conclusion (see section 2.2).
Some vague descriptions in the manuscript may mislead you regarding the purpose and conclusions of our research. We owe you a apology. We have revised the Abstract, Introduction and Conclusion of the manuscript to emphasize that the focus of our work is to compare which of these descriptive methods and machine learning methods are more suitable for multi-classification of drugs. Our contribution is also to propose some improvements in description methods and classification methods to improve the accuracy of multi-classification of drugs.
We appreciate reviewer’s attention and comments on our work, which make our revision more clear and complete.

This manuscript is a resubmission of an earlier submission. The following is a list of the peer review reports and author responses from that submission.
Round 1
Reviewer 1 Report
Zhong et al., in their study mentioned therapeutic based drug screening as early drug discovery. I have few comments to make:
- The study lack proper description of all the software and algorithms used in the study.
- Does this algorithm consider chirality or optical isomers since its an important parameter for drug activity?
- All the method description needs to be in past tense.
- The study lacks the future applications and potential of this study in the benefit of combating the disease.
- The study has lots of flaws, Since most of the important drugs have more than one therapeutic use, and some diseases like cancers are group of diseases, so how do you justify validity of this study by excluding potential drug molecules?
Reviewer 2 Report
It can be accepted after some minor English corrections
Reviewer 3 Report
Under the title " Small molecular drug screening based on clinical therapeutic effect " the authors
describe a computed study with 1019 drugs. They were collected with a specific therapeutic effect (analgetics)
from various data bases and some published documents.
The drugs were grouped following the Figure 2 which shows the overall work flow.
To obtain the data set, hand selction was critical, i.e.
"carefully checking therapeutic use and mechanism" for the " drugs with analgesic effect contain analgesics,
and several anesthetics and antipsychotics with analgesic effects, where several drugs only have local anesthesia or muscle relaxation.
Those drugs cannot inhibit the expression of cyclooxygenase like most analgesics,
just can be used to relieve pain or paralyze nerves by other analgesic mechanisms. ".
Problem 1:
The pharmacologic term for the studied " specific therapeutic effect " is in need of a specific target
biomolecule, e.g. COX-2 for the antiinflammatory action mecanism. But it is not "expressis verbis" =
stated explicitly. It is not told , neither in Abstract no Introduction, which is the target receptor.
But this reviewer, cannot find it in the text.
How such mechanism of action are linked to experiments, is illustrated here, for example: histology for events of
antitubulin agents: " Efficacy in murine preclinical models correlated semiquantitatively, with CAIX
expression levels as determined by immunohistochemistry and ELISA "
Cited from [Therapeutic Discovery, Mol Cancer Ther; 11(2) February 2012;
"Therapeutic Mechanism and Efficacy of the Antibody–Drug
Conjugate BAY 79-4620 Targeting Human Carbonic
Anhydrase 9"; Heike M. Petrul, Christoph A. Schatz, Charlotte C. Kopitz, Lila Adnane, Timothy J. McCabe, Pamela Trail,
Sha Ha, Yong S. Chang, Andrei Voznesensky, Gerald Ranges, and Paul P. Tamburini].
Problem 2:
For the Abstract : What is the definition of " therapeutic mechanism information from clinic.... " ? and,
how it is used in the study? This idea should be graphically illustrated or explained by words in the Legend of Figure 2.
Is this term equal to "molecular mechanism of action" ? And only twice the term appears in the text. but it should be
described and literature cited, because this reviewer, cannot find it in the text.
Problem 3:
The critical point to assess the statistical significance of results is given in
section 2.2.3 (random sampling) , the KS or " Kennard-Stone method " is applied to assign training sets and the test sets based on difference between samples.
This is very complicated assigment. Is it really necessary? Why not randomly assign test and training sets?
And repeat it ? Also, why not use LEAVE ONE OUT , or BOOTSTRAPING ?
Problem 4:
The collected drugs may be do not share the same target receptor, here COX ?
For example, see cases like
ketamine which "exhibits a rapid and persistent effect against PTSD,
though the underlying molecular mechanism remains to be clarified. ", found in
[ Epub 2021 Apr 13. "Ketamine for post-traumatic stress disorders and it's possible therapeutic mechanism" ;
Muhammad Asim 1; Bing Wang 2; Bo Hao 3; Xiaoguang Wang; PMID: 33862176 DOI: 10.1016/j.neuint.2021.105044. ]
Problem 5:
Line 161 :
Secction 2.1 informs about molecular sets and how to calculate descriptors.
" Both molecular descriptor and molecular fingerprint are quantitative data converted
from structural information."
Line 163 to Line 166:
" The SMILES, as one of molecular structure representations, is often used as input for computational programs
to calculate the descriptors of molecules.
In order to achieve better representation for drug molecular structure, different forms of properties are
acquired and used as feature data to classify drugs. Two types of the molecular descriptors, ... " .
Of note, these two types are " the combinatorial descriptor set" and " Mordred descriptor group, ... incluing multiple sets of descriptors, is served as another descriptor set."
Here the study design is at stake, the problem is that the entire structural information of the (hand-)selected molecules is not the correct way to proceed.
NOT all parts of the structure are informative about how they CORRELATE with THERAPEUTIC MECHANISM !
Only those parts of the structures are informative which INTERACT with TARGET.
The other parts only put "data noice" to it, and the statistical outcome and the complicated sampling mehtod, KS, etc,
only reflect, that here something may go wrong.
The pharmacophore would be a "BETTER" way = see what the authors tried to do : " ...
to achieve better representation for drug molecular structure, ... ",
but they did not see that only parts are RELEVANT to be considered ESSENTIAL for the statistical analysis.
Also, a wast body of literature exists in case of COX, COX-2 analgesics and molecular action mechanism,
and many "snapshots" between ligand drugs and target receptor at atomic scale exist.
So, here the authors intent to correlate vaguely by taking the entire molecules, while the
Medicinal Chemistry science already is a step further just to take the pharmacophore model.
Also concepts of prodrug, active metabolites and so on are not included in the DATA SETS, all of which
may heavily modify or contribute to activity.
So, the work,it is not conclusive in this form.
Reviewer 4 Report
In this work, the authors used a multi-classification method based on Dempster-Shafer (DS) evidence reasoning theory to predict clinical therapeutic effect of unknown drugs. They selected random forest (RF), adaptive boosting trees (ABT), support vector machine (SVM), logistic regression (LR), and linear discriminant analysis (LDA) as the single classifier for fusion to study a four drugs molecular sets (seven types of drugs were collected with two or even three therapeutic uses). They predicted probabilities of the five classifiers that are fused into final discriminant probability. The Kennard-Stone (KS) division method was used to select the model with good performance to give more reliable information for potential effect of unknown drugs. They verified the accuracy of the obtained model by an external validation set. The autohrs concluded that the obtained model not only performs well but also is able to identify multiple use of drugs, which will provide more realistic prediction for unknown drugs and discover potential new usage for old drug.
I have some comments and questions:
- a) To avoid confusion with references, on page 7, line 290, I suggest not using square brackets in the paragraph "... with a range of values ​​of [0, 1]".
- b) The authors suggested that this research will provide valuable information for drug discovery and classification. Is there experimental data to corroborate the suggestion?.
- c) The authors studied 1019 drug molecules, but they only mentioned rifampicin and celecoxibion, what are the other molecules?.
The manuscript is interesting, well written and easy to read. I think it is appropriate for the journal.